A reappraisal of Polyptychodon (Plesiosauria) from the Cretaceous of England

Madzia Daniel daniel.madzia@gmail.com
Institute of Paleobiology, Polish Academy of Sciences , Warsaw , Poland
Young Mark
Electronic publication date: 2016 May 10
Publication date: 2016
Volume: 4
Electronic Location ID: e1998
Received 2016 Feb 25; Accepted 2016 Apr 11
Copyright: ©2016 Madzia
Copyright year: 2016
Copyright holder: Madzia
License: This is an open access article distributed under the terms of the Creative Commons Attribution License, which permits unrestricted use, distribution, reproduction and adaptation in any medium and for any purpose provided that it is properly attributed. For attribution, the original author(s), title, publication source (PeerJ) and either DOI or URL of the article must be cited.
License URL: https://creativecommons.org/licenses/by/4.0/

Keywords: Polyptychodon, Plesiosauria, Pliosauridae, Cretaceous, Teeth, Cambridge Greensand, England

Funding: National Science Centre (Poland) DEC-2012/05/B/ST10/00710 The study was funded by the National Science Centre (Poland) grant DEC-2012/05/B/ST10/00710 to Marcin Machalski (Institute of Paleobiology, Polish Academy of Sciences). The funder had no role in study design, data collection and analysis, decision to publish, or preparation of the manuscript.

==============================
Pliosauridae is a globally distributed clade of aquatic predatory amniotes whose fossil record spans from the Lower Jurassic to the Upper Cretaceous. However, the knowledge of pliosaurid interrelationships remains limited. In part, this is a consequence of a few key taxa awaiting detailed reassessment. Among them, the taxon Polyptychodon is of special importance. It was established on isolated teeth from the mid-Cretaceous strata of East and South East England and subsequently associated with numerous finds of near-cosmopolitan distribution. Here the taxon is reassessed based on the original dental material from England, with special focus on a large collection of late Albian material from the Cambridge Greensand near Cambridge. The dental material is reviewed here from historical and stratigraphic perspective, described in detail, and discussed in terms of its diagnostic nature. The considerable morphological variability observed in the teeth attributed to Polyptychodon, together with a wide stratigraphic range of the ascribed material, possibly exceeding 35 Ma (early Aptian to ?middle Santonian), suggests that the taxon is based on a multispecies assemblage, possibly incorporating members of different plesiosaur clades. Due to the absence of any autapomorphic characters or unique character combinations in the original material, Polyptychodon interruptus, the type species of Polyptychodon, is considered nomen dubium. From a global perspective, Polyptychodon is viewed as a wastebasket taxon whose material originating from different localities should be reconsidered separately.

Introduction

Pliosaurid plesiosaurs were highly successful aquatic predatory amniotes that represented significant components of Mesozoic marine ecosystems. Along with xenopsarians (i.e., elasmosaurids and leptocleidians) and a single cryptoclidid species Abyssosaurus nataliae Berezin, 2011, pliosaurids are the only known plesiosaurs that crossed the Jurassic-Cretaceous boundary. All other plesiosaurs became extinct until the end of the Jurassic (Benson & Druckenmiller, 2014).

Whereas the Middle to Late Jurassic pliosaurid record is relatively abundant (e.g., Knutsen, 2012; Knutsen, Druckenmiller & Hurum, 2012; Benson et al., 2013), the Cretaceous pliosaurids are represented by only a few taxa, most or all of which belong to a single lineage named Brachaucheninae (Benson et al., 2013; Benson & Druckenmiller, 2014; Cau & Fanti, 2015; Zverkov, 2015; Fischer et al., 2015). Unfortunately, our knowledge of brachauchenine origins, interrelationships, paleoecology, and paleobiogeography is rather poor. In part, this might be due to scarce fossil material of the oldest known members of this clade (Hampe, 2005; Fischer et al., 2015) and low taxic diversity in the Early Cretaceous (Benson et al., 2010). Other reasons might include general absence of detailed taxonomic assessments of the Cretaceous pliosaurid record.

The best known brachauchenines are Brachauchenius lucasi Williston, 1903 from the lower to lower middle Turonian of Kansas, USA (see also Schumacher & Everhart, 2005), Megacephalosaurus eulerti Schumacher, Carpenter & Everhart, 2013 from the lower middle Turonian of Kansas, USA, and Kronosaurus queenslandicus Longman, 1924 from the Aptian-Albian of Queensland, Australia (see Kear, 2003). Due to their reasonably complete nature, these three taxa usually represent the key brachauchenines while inferring pliosaurid phylogenetic relationships (e.g., Ketchum & Benson, 2010; Benson et al., 2013; Benson & Druckenmiller, 2014).

Still, none of these taxa is considered to be as widely distributed geographically as the purported brachauchenine Polyptychodon Owen, 1841a. Although originally described from the Cretaceous strata of East and South East England (e.g., Owen, 1841c; Owen, 1851), the fossils ascribed to this taxon, either tentatively or with certainty, have been noted from numerous localities across the world. In Europe, the material associated with Polyptychodon originates from various localities in the Czech Republic (e.g., Kear et al., 2014), France (Barrois, 1875; Buffetaut et al., 2005), Germany (Wagner, 1853; Sachs, 2000; Sachs et al., 2016), Italy (Papazzoni, 2003), Poland (Marcinowski & Radwański, 1983; Bardet, Fischer & Machalski, 2016), Russia (Kiprijanoff, 1883; Fischer et al., 2016), and Ukraine (Schloenbach, 1868). Polyptychodon was also recorded from Texas and South Dakota in the United States (Welles & Slaughter, 1963; VonLoh & Bell, 1998; respectively), Hokkaido in Japan (Obata, Hasegawa & Otsuka, 1972; Echizenya, 2011), and Santa Cruz Province in Argentina (Ameghino, 1893). However, it is important to note that the last two occurrences of this taxon were subsequently questioned due to unknown diagnostic nature of the original material of P. interruptus and provisionally referred to as Pliosauroidea indet. and Plesiosauria indet. (Sato et al., 2012; O’Gorman & Varela, 2010; respectively).

Other occurrences, such as Polyptychodon rugosus (Emmons, 1858) and Plesiosaurus (Polyptychodon) mexicanus (Wieland, 1910) are of marginal importance as they were conclusively shown to represent members of different clades (Baird & Horner, 1979; Buchy, 2008; respectively). Additional material associated with Polyptychodon was described by Deslongchamps (in Lennier, 1870) and Lennier (1887). Deslongchamps (in Lennier, 1870) established a new species, Polyptychodon archiaci, on a lower jaw (MNHN cat.24.1) from the Kimmeridgian (Upper Jurassic) of Le Havre (France). The material was first figured by Fischer (1869) as Pliosaurus grandis and later considered to be referable to Stretosaurus macromerus by Tarlo (1960) and to Pliosaurus brachyspondylus by Bardet, Mazin & Martin (1993).

Discussions of the taxonomic validity of Polyptychodon have previously been published (Welles, 1962; Welles & Slaughter, 1963; Albright, Gillette & Titus, 2007; Schumacher, 2008; Schumacher, Carpenter & Everhart, 2013; Angst & Bardet, 2016). Nevertheless, they were often brief, and although inclined to treat Polyptychodon as problematic or even invalid (e.g., Welles, 1962), the matter of its taxonomic validity remained open.

The aim of this study is to reassess the taxonomic validity of Polyptychodon from the Cretaceous of East and South East England. It is based especially on extensive collection of late Albian teeth from the Cambridge Greensand of East England, housed at the Sedgwick Museum of Earth Sciences, Cambridge, that was supplemented with additional isolated teeth from the Lower and Upper Cretaceous of England, deposited at the same institution. These collections were selected because the name-bearing specimens (Owen, 1841a; Owen, 1841b; Owen, 1841c; Owen, 1851) were not found and are possibly lost. The collection of the Sedgwick Museum of Earth Sciences is considered to be a representative sample of Polyptychodon, most closely corresponding to the name-bearing material in terms of morphology and stratigraphic provenance. It also includes some specimens initially described by Owen (1851; e.g., CAMSM B 57400). Thus, it might be considered a part of the original material attributed to Polyptychodon and an appropriate basis for detailed assessment of this historically important taxon.

The postcranial material ascribed to Polyptychodon was not evaluated in the present study because the taxon is established on isolated teeth. Unless the tooth material is properly assessed, its connections with particular postcranial remains discovered in the contemporary strata are impossible to be settled with certainty.

The results of the present study are expected to have an impact on our understanding of the dental morphology of mid-Cretaceous robust-toothed plesiosaurs and enable to appraise the taxonomic composition of abundant collections of isolated dental elements (such as, for example, the one from the mid-Cretaceous condensed sedimentary succession at Annopol, Poland; Marcinowski & Radwański, 1983; Bardet, Fischer & Machalski, 2016).

Material and Methods

The reappraisal of Polyptychodon is divided into three parts. The first part presents the historical background of the initial discoveries. The second part consists of a discussion of their stratigraphic settings. The third part includes an assessment of the morphological variability in the teeth regarded as representing Polyptychodon interruptus. It is based on direct examination of 135 isolated teeth housed at the Sedgwick Museum of Earth Sciences, University of Cambridge, Cambridge.

Photographs were taken using digital single-lens reflex camera Nikon D1X.

Tooth anatomical orientation. The terminology of anatomical orientation follows Smith & Dodson (2003): apical, toward the apices of the tooth crown or the tooth base; basal, toward the cervix dentis; distal, away from the tip of the snout; labial, toward the lips; lingual, toward the tongue; mesial, toward the tip of the snout (see Fig. 1).

Figure 1 Tooth anatomical orientation in idealized plesiosaur tooth.

(A) plesiosaur tooth crown in labial view; (B) apical view of plesiosaur tooth crown. Pictures roughly based on CAMSM B 57378.

Historical Background

Due to the absence of internationally accepted rules governing zoological nomenclature during the 19th century, many taxa introduced at that time lacked adequate descriptions. Polyptychodon has been frequently cited taxon since its initial establishment in 1841 by famous British vertebrate paleontologist Richard Owen, and besides being often viewed as problematic (O’Gorman & Varela, 2010; Sato et al., 2012), it still constitutes an important reference taxon while evaluating systematic affinities of plesiosaur remains accompanied with robust teeth (e.g., Bardet, Fischer & Machalski, 2016). As such, it is necessary to review the history and geological context of the material originally ascribed to Polyptychodon in detail.

The review consists of two parts. The first part deals with the material that represents the name-bearing specimens. The second part deals with additional original English material ascribed to Polyptychodon. These specimens are reviewed with regard to their county of discovery.

Name-bearing specimens

The name Polyptychodon first appeared in 1841’s “Part II: Dental System of Reptiles” of Richard Owen’s monumental work “Odontography; or, a Treatise on the Comparative Anatomy of the Teeth; their Physiological Relations, Mode of Development, and Microscopic Structure, in the Vertebrate Animals” published between 1840 and 1845. However, the information regarding the taxon published by Owen (1841a) is limited to the mention of the name Polyptychodon and a brief account of the outer surface of its tooth crowns, only described as possessing “many narrow ridges” (Owen, 1841a; p. 181). Two species were introduced, P. continuus and P. interruptus (Owen, 1841b; p. 19), each based on a single tooth crown illustrated on the Plate 72; Figs. 3 and 4, respectively (see Fig. 2). No additional discussion was provided.

In the same year, Owen (1841c; p. 156–157) published a more precise description of the material:

“A large species of Saurian is indicated by thick conical teeth, having the general character of those of the Crocodile, but distinguished by numerous, closely-set, longitudinal ridges, which are continued, of nearly equal length, to within 2 lines of the apex of the crown. [...] The tooth of the Polyptychodon is slightly and regularly curved, and invested with a moderately thick layer of enamel, of which substance the ridges are composed, the surface of the outermost layer of dentine is being smooth. A tooth of this reptile from the lower greensand (Kentish-rag quarries) near Maidstone, in the collection of Mr. Benstead of that town, has a crown 3 inches long, and 1 inch 4 lines across the base. The compact dentine is resolved by decomposition into a series of superimposed thin hollow cones, and the short and wide conical pulp-cavity is confined to the base of fang.”

It is beyond doubt that the tooth mentioned by Owen (1841c) is the incomplete tooth crown named as Polyptychodon continuus, that appeared on Fig. 3 of the Pl. 72 (Owen, 1841b). The same tooth was illustrated by Owen (1851) on Table. XIV, Figs. 5 and 6, and labeled as “From the Kentish Rag, Green-sand Formation, near Maidstone. In the Collection of J. Bensted, Esq.” Owen (1851; p. 47) also provided more information on the provenance of the tooth:

“The first evidence of this species was a single tooth, which was discovered by W. H. Bensted, Esq., of Rock Hall, near Maidstone, September 16th, 1834, in what is called the ‘Trigonia-stratum’ of Shanklin Sand, in the Kentish Rag Quarries near that town, this stratum being a member of the Lower Green-sand Formation.”

The tooth named Polyptychodon interruptus (Owen, 1841b; Pl. 72, Fig. 4) has never been described. Moreover, Owen never specified the precise locality where it was discovered. Welles (1962: p. 61) suggested that the name-bearing tooth crown was redrawn by Owen (1851; Table. XIV, Figs. 1 and 2). If true, the specimen could have originated from the “Chalk of Sussex” (see also ‘Discussion’). Such provenance might also ensue from Owen’s later writings where he mentioned that “the genus [Polyptychodon was] founded, in 1841, on certain large detached teeth from the Cretaceous beds of Kent and Sussex” (Owen, 1860; p. 262).

Figure 2 Name-bearing specimens of (A) Polyptychodon continuus from the Aptian of the Hythe Formation (presumbly from the labiodistal view) and (B, C) P. interruptus most likely from the Upper Cretaceous of “Sussex” (from the [B] labial and [C] lingual views).

Vectored from Owen (1841b; Plate 72; Figs. 3 and 4, respectively) using Vector Magic (Cedar Lake Ventures, Inc.).

Nevertheless, when discussing the provenance of P. interruptus, Owen (1851; p. 55–56) noted:

“The majority of the specimens of the teeth of this species [P. interruptus—DM] have been found in the middle and lower Chalk or Chalk-marl: one large tooth of this species has been discovered by the Rev. Peter Brodie, M.A. F.G.S., in the upper Green-sand at Barnwell, near Cambridge, and a few other specimens have been obtained by James Carter, Esq. from the Green-sand of another locality, near Cambridge.”

Also, Owen (1851; p. 56) provided a brief comparison of P. continuus and P. interruptus and pointed out the differences between both taxa:

“The general shape of the crown [of P. interruptus—DM] agrees with that of the Polyptychodon continuus; the differences is shown by the greater proportion of the ridges which stop short of the apex of the crown, especially on the convex side of the tooth.”

Owen (1851; p. 56) further continued with a more detailed description of a P. interruptus tooth found near Lewes (East Sussex), which he evidently considered a typical representative of that species:

“Around the entire basal part of the crown the ridges are close together: their interspaces are only clefts that separate them. On the concave side of the tooth a large proportion of the ridges extend nearly to the apex, as is shown in Table. XI, Fig. 1; but on the convex side a greater number extend only one third or two thirds towards the apex, these shorter ridges alternating with the longer ones, between which, therefore, at the apical part of the tooth there are intervals of the flat tracts of enamel. The apex of the tooth is rather obtuse. On one side of the crown there is a long ridge, towards which contiguous shorter ones have a convergent inclination. The long fang of the tooth is covered by a layer of smooth cement. The dentine is compact, and corresponds in microscopic structure with that of the crocodile’s teeth.”

As it is apparent from Owen (1851; p. 56), his use of the terms “convex” and “concave”, while referring to tooth sides, roughly correspond with “mesiolabial” and “linguodistal” sides, respectively.

Additional original material from England

Following the early discoveries of Polyptychodon published by Owen (1841a), Owen (1841b) and Owen (1851), other mid-Cretaceous specimens were ascribed to this taxon. These remains were unearthed from the deposits of East and South East England; in the counties of Cambridgeshire, East Sussex, Kent, Somerset(?), Surrey, and West Sussex.

The documented material is summarized in Table 1.

Table 1 Original material attributed to Polyptychodon.

Material	First attributed to Polyptychodon by	Original identification	Locality and year of discovery	Initial description of the provenance	Modern stratigraphic terminology	Stage	
Tootha	Owen (1841b; p. 19) and Owen (1841b: Pl. 72, Fig. 3)	P. continuus	Kentish Rag Quarries near Maidstone (Kent); 1834	Lower Greensand Formation, ‘Trigonia-stratum’ of Shanklin Sand	Hythe Formation	Aptian	
Toothb	Owen (1841b; p. 19) and Owen (1841b: Pl. 72, Fig. 4)	P. interruptus	Sussex?	Chalk?	Chalk Group?	? (Upper Cretaceous)	
Incomplete postcranial skeletonc	Owen (1841c; pp. 157–159)	P. continuus	Hythe (Kent); 1840	Lower Greensand	Hythe Formation?	Aptian	
Tooth	Owen (1850; p. 378, and Table XXXVII: Figs. 16 and 17)	P. interruptus	Valmer, Lewes (East Sussex); ?	Chalk	Newhaven Chalk Formation?	Santonian–lower Campanian	
Fragmentary lower jaw	Owen (1850; p. 379, and Table XXXVIII: Fig. 3)	P. interruptus	Unknown locality (Kent); ?	Lower Chalk	West Melbury Marly Chalk or the Zig Zag Chalk Formation?	Cenomanian?	
Tooth	Owen (1851; p. 47, and Table XIV: Fig. 4)	P. continuus	Unknown locality (Kent); ?	Chalk	Chalk Group	? (Upper Cretaceous)	
Unknown number of teeth	Owen (1851; pp. 55–56, and probably Table X: Figs. 8 and 9)	P. interruptus	Barnwell and “another locality” (Cambridgeshire); ?	Middle and lower Chalk or Chalk-marld	Cambridge Greensand Member of the West Melbury Marly Chalk Fm?	Cenomaniane	
Eight teeth	Owen (1851; p. 56, and Table XI: Figs. 1–7)	P. interruptus	Lewes (East Sussex); 1847	Lower Chalk	Glauconitic Marl Member of the West Melbury Marly Chalk Formation	Cenomanian	
Tooth	Owen (1851; p. 57, and Table XI: Fig. 8)	P. continuus?	Houghton, near Arundel (West Sussex); 1850	Lower Chalk	West Melbury Marly Chalk or the Zig Zag Chalk Formation	Cenomanian	
Fragmentary lower jaw	Owen (1851; p. 58, and Table XVI)	Polyptychodon	Burham Chalk-pit (Kent); ?	Chalk of Kent	Zig Zag Chalk Formation (alternatively the upper section of the West Melbury Marly Chalk Fm or the lowest members of the Holywell Nodular Chalk Fm)	Cenomanian–lower Turonian	
Unknown number of teeth	Owen (1851; p. 58, and Table X: Figs. 8 and 9)	P. interruptus	Cambridge (Cambridgeshire); ?	Upper Greensand	Cambridge Greensand Member of the West Melbury Marly Chalk Formation	Lower Cenomaniane	
Incomplete cranial material	Owen (1860a; p. 262)	P. interruptus	Dorking (Surrey); ?	Lower Chalk	West Melbury Marly Chalk, Zig Zag Chalk, Holywell Nodular Chalk or the New Pit Chalk Formation	Cenomanian–middle Turonian	
Teeth and vertebrae	Owen (1860a; p. 262)	Polyptychodon	Cambridge (Cambridgeshire); ?	Upper Greensand	Cambridge Greensand Member of the West Melbury Marly Chalk Formation	Lower Cenomaniane	
Unspecified limb bones	Owen (1860a; p. 263)	Polyptychodon	Cambridge (Cambridgeshire); ?	Greensand beds	Cambridge Greensand Member of the West Melbury Marly Chalk Formation	Lower Cenomaniane	
Phalanges	Owen (1860a; p. 263)	Polyptychodon	Unknown locality (Kent); ?	Chalk	Chalk Group	? (Upper Cretaceous)	
Unspecified number of teeth	Owen (1860a; p. 263)	Polyptychodon	Fromef (Somerset); ?	Chalk	Chalk Group	? (Upper Cretaceous)	
Dorsal vertebra	Owen (1861; p. 23)	P. interruptus	Cambridge (Cambridgeshire); ?	Upper Greensand	Cambridge Greensand Member of the West Melbury Marly Chalk Formation	Lower Cenomaniane	
Fragmentary ribs	Owen (1861; p. 23; Table V: Fig. 3)	P. interruptus	Cambridge (Cambridgeshire); ?	Upper Greensand	Cambridge Greensand Member of the West Melbury Marly Chalk Formation	Lower Cenomaniane	
Three teethg	Seeley (1869; p. 3)	P. interruptus	Halling (Kent); ?	Chalk	Plenus Marls Member of the Holywell Nodular Chalk Formation?	Upper Cenomanian	
Two teethg	Seeley (1869; p. 3)	P. interruptus	Offhamf (Kent); ?	Chalk	Chalk Group	? (Upper Cretaceous)	
Three teethg	Seeley (1869; p. 3)	P. interruptus	Cherry Hinton (Cambridgeshire); ?	Chalk	West Melbury Marly Chalk Formation or the Zig Zag Chalk Formation	Cenomanian	
103 teethg,h	Seeley (1869; pp. xviii, 45)	P. interruptus	Cambridge (Cambridgeshire); ?	Upper Greensand	Cambridge Greensand Member of the West Melbury Marly Chalk Formation	Lower Cenomaniane	
15 cervical and dorsal vertebrae	Seeley (1869; p. xviii, 45)	P. interruptus	Huntingdon Road (Cambridgeshire); ?	Upper Greensand	Cambridge Greensand Member of the West Melbury Marly Chalk Formation	Lower Cenomaniane	
Notes.

a Name-bearing specimen of P. continuus.

b Name-bearing specimen of P. interruptus.

c Today considered to be an indeterminate macronarian sauropod (Mannion et al., 2013).

d Although Owen (1851) also mentioned “middle Chalk”, he only discussed the “Greensand” near Cambridge which most likely means the strata belonging to the Cambridge Greensand Member.

e Majority of the material from the Cambridge Greensand Member is derived from the upper Albian Gault Formation.

f The provenance might be incorrect (see text).

g Described in the present study

h Some of the teeth have previously been published by Owen (1851).

Some tooth material historically classified as Polyptychodon is on display in museums and labeled as belonging to this taxon. However, many of these specimens have never been described or figured. Also, they were often unearthed from the same strata as the teeth originally mentioned by Owen (1841a), Owen (1841b) and Owen (1851), with most of them being unearthed from the Cambridge Greensand Member or generally in the vicinity of Cambridge (e.g., a large part of the tooth material at the collection of NHMUK (NHM 2014); all tooth specimens at the New Walk Museum (M Evans, pers. comm., 2016)). Thus, this paper does not mention all dental material attributed to Polyptychodon. Rather, effort was made to cite especially its earliest finds that served as the basis for further assignments.

Cambridgeshire. Most of the specimens described from East England are currently housed in the Sedgwick Museum of Earth Sciences of the University of Cambridge. When known, their localities and current catalog numbers are provided in the present paragraph. The morphology is assessed in a separate section. Owen (1851; p. 55–56) commented on an unspecified number of isolated teeth from the “middle and lower Chalk or Chalk-marl” near Cambridge. One tooth originated from Barnwell and “a few other specimens” from “another” site close to Cambridge. Later on, Owen (1851; p. 58) noted that tooth crowns of both species of Polyptychodon, purportedly illustrated on his Table X (Figs. 8 and 9), were discovered in the “Upper Greensand” of Cambridge and at Horningsea (called “Horn-sea” by Owen) in Cambridgeshire. While it seems likely that some of these teeth from the Greensand mentioned on p. 58 were already among the “other specimens” from “another locality” listed earlier by Owen (they were obtained by the same person, James Carter), the material from Horningsea remained undiscussed. Unfortunately, both figures mentioned by Owen (1851; p. 58), i.e., Figs. 8 and 9 on the Table X, illustrate a single tooth regarded as P. interruptus (CAMSM B 57400). Thus, it remains unknown whether Owen meant to say that P. continuus was also discovered in Cambridge or at Horningsea.

Additional material was mentioned by Owen (1860). It included “teeth of Polyptychodon, with plesiosauroid vertebrae of the same proportional magnitude” (p. 262) and “[p]ortions of large limb-bones, without medullary cavity and of plesiosauroid shape” (p. 263). In both cases the material was discovered in the “Greensand”. The teeth and vertebrae, then, were stated to be housed “in the Woodwardian Museum [the Sedgwick Museum—DM]” (Owen, 1860; p. 262). The vertebrae are probably the same as those described by Owen (1861; p. 22–24) from the “Upper Greensand” of Haslingfield, and illustrated on the Table V and VI. Their current catalog numbers are CAMSM B 57275, B 57279, and B 57280.

In addition, Owen (1861; p. 23) mentioned “the centrum of a dorsal vertebra” and “portions of ribs” from the “Upper Green-sand” of Cambridgeshire. The dorsal vertebra might not be cataloged but there are three vertebrae described as cervical in the CAMSM collections (CAMSM B 57276–78) and a rib fragment (CAMSM B 57281).

The most extensive list of the Polyptychodon material from Cambridgeshire was published by Seeley (1869). Except for several specimens described previously by Owen in 1851 and 1861, most of the fossils have not been explicitly mentioned before, although it is probable that Owen was familiar with their existence or even considered them when discussing tooth material from the “Upper Greensand” of Cambridgeshire. The material contains three teeth from the “Lower Chalk” of Cherry Hinton (CAMSM B 20624–26), seven cervical and nine dorsal vertebrae from the “Cambridge Upper Greensand” of Huntingdon Road (CAMSM B 57385–99), three vertebrae and a rib fragment (CAMSM B 57276–78, B 57281), 103 isolated teeth (CAMSM B57282–384), unspecified number of “chiefly duplicate” teeth, and partial femur (CAMSM B 57851) from the “Cambridge Upper Greensand” of unknown localities near Cambridge.

The reason why Seeley (1869; p. 45) listed some of the specimens, such us the high number of isolated teeth (CAMSM B57282–384), in separate rows remains unknown. It might have been simply due to space reasons as Seeley’s (1869) index is structured according to the positions of the specimens in particular museum drawers. However, the separation might also reflect distinct associations or different localities of discovery.

East Sussex. Nine teeth were published from the “Chalk” in the vicinity of Lewes. The first material to be mentioned consisted of a single tooth that was found “near Valmer in cutting the Lewes railway” (Owen, 1850; p. 378). Additional eight isolated teeth, with at least two of them being found in a close association (see Owen, 1851; Table XI, Fig. 3), “were discovered […] in the lower bed of Chalk-marl, just above the Green-sand, in the vicinity of that town” (Owen, 1851; p. 56).

Kent. The Polyptychodon fossil remains from the deposits of the county of Kent is the second richest collection after the Cambridge Greensand material. The first discovery of Polyptychodon from Kent was the isolated tooth found near Maidstone that represents the name-bearing specimen of Polyptychodon continuus. Additional material purportedly conspecific with P. continuus was found in the Lower Greensand at Hythe and consisted of an incomplete postcranial skeleton first discussed by Owen (1841c) and interpreted as including “portions of coracoid, humerus, and ulna, of the iliac, ischial, and pubic bones, a large proportion of the shaft of a femur, parts of a tibia and fibula, and several metatarsal bones, four of which exhibit their proximal articular surfaces” (Owen, 1851: p. 48). Owen (1851; p. 47) tentatively assigned these fossil remains to P. continuus “on account of the identity of the Formation [i.e., ‘Lower Greensand’—DM] in which they were discovered, with that of the tooth of Polyptychodon continuus [i.e., the name-bearing specimen of P. continuus from the Lower Greensand near Maidstone—DM] and because no other teeth have as yet been found in the Cretaceous Series to which the fossils in question could be referred”. Later on, however, Owen (1849–84a; p. ix) noticed that the remains belong to a dinosaur rather than a sauropterygian and named them Dinodocus mackesoni. This material is currently regarded as belonging to an indeterminate macronarian sauropod (NHMUK 14695; Mannion et al., 2013).

The Kent material also includes two fragmentary lower jaws. The first one was discovered in the “Lower Chalk” of an unspecified locality and ascribed to P. interruptus (Owen, 1850; p. 379). The second one, on the other hand, originated from “the Burham Chalk-pit” (Owen, 1851; p. 58). This fragment, which was interpreted by Owen (1851; p. 58) as the “anterior end of the left ramus”, did not preserve any teeth. Thus, Owen (1851) did not attempt to identify it to species and simply labeled it as Polyptychodon.

While discussing newly identified material of P. continuus, Owen (1851; p. 47) stated: “In the Collection of Henry Catt, Esq., of Brighton, is preserved the crown of a nearly equally fine specimen of Polyptychodon continuus, from the Chalk of Sussex: this specimen is figured of the natural size in Table XIV, Fig. 4.” However, description of the Fig. 4 (Table XIV) of Owen (1851) states: “Crown of the tooth of Polyptychodon continuus. From the Chalk of Kent. In the collection of H. W. Taylor, Esq., of Brixton Hill.” Interestingly, the same provenance, as in the case of the above-mentioned Catt’s tooth, was published by Owen (1850; p. 378) for a tooth from the “Chalk” near Valmer. This specimen was first depicted on Table XXXVII (Figs. 16 and 17) and later mentioned by Owen (1851; p. 57, and Table IX, Figs. 11 and 12). In both cases, Owen interpreted the Valmer tooth as Polyptychodon interruptus. Thus, while referring to Fig. 4 on the Table XIV, Owen (1851: p. 47) probably confused the discoverer and the provenance of a tooth originating, in fact, from Kent.

Additional material attributed to Polyptychodon included an incomplete “large Plesiosauroid paddle, from the Chalk of Kent” (Owen, 1860; p. 263). This material had previously been interpreted as belonging to Plesiosaurus (see Owen (1849–84b; Plate 30) and Owen (1851; Table XVII)).

Polyptychodon was described from the “Chalk” of Kent also by Seeley (1869; p. 3) who reported on three teeth found in Halling (CAMSM B 20619–21) and two others from Offham (CAMSM B 20622, 23). This material is described herein in a separate section.

Somerset. Fossil material from Somerset consists of an unspecified number of teeth “in the collection of W. Harris, Esq., F.G.S.”, supposedly unearthed “from a chalky deposit with greenish granules, in a tunnel of the railway near Frome” Owen (1860; p. 263). Unfortunately, this description is most likely imprecise or incorrect. The closest “railway tunnel” situated within the Upper Cretaceous strata is located some 8 km east of Frome, near the village of Upton Scudamore, in the county of Wiltshire. These strata belong to the West Melbury Marly Chalk Formation (P Hopson, pers. comm., 2015).

Surrey. While preparing Supplement III to his “Monograph” (Owen, 1861), Owen was shown incomplete cranial remains found by George Cubitt. The material consisted of isolated tooth, jaw fragment, and partial skull, including incomplete premaxillae, the parietal, and the squamosals (Owen, 1861; Table IV). Owen (1861) noted that the remains were “discovered in cutting a railway tunnel through the Chalk formations near Frome, Somersetshire” (p. 20), but this locality was most likely confused with that of other specimens attributed to Polyptychodon; teeth mentioned by Owen (1860; p. 263). When Owen briefly commented on the cranial material for the first time in 1860, it was introduced as being from “the Lower Chalk at Dorking” (p. 262). Since (1) the discoverer George Cubitt became 1st Baron Ashcombe of Dorking in 1892, (2) the specimen is currently housed at the Dorking & District Museum (under the catalog number DOKDM G/1–2), in Dorking, Surrey, and (3) the “Chalk” was quarried in the vicinity of the town, the originally mentioned provenance seems to be more likely (see also Benson et al., 2013). According to Benson et al. (2013), the specimen should be referred to as Brachauchenius indet.

West Sussex. Owen (1851; p. 57–58) described an isolated tooth from the Lower Chalk at Houghton, near Arundel. According to Owen (1851; p. 58), the tooth ridges differed from other teeth he described:

“One or two of the long ridges […] are more than usually prominent, and most of the shorter ones are fainter than usual […]” (p. 58).

Still, Owen (1851) considered the differences being most likely the result of individual variability and listed the specimen as “Polyptychodon continuus (?)” (Owen, 1851; Table XI, Fig. 8).

Stratigraphic Settings

There have been considerable changes in the nomenclature of the Cretaceous lithostratigraphic units of England, since Owen (1841a), Owen (1841b), Owen (1841c), Owen (1850) and Owen (1851) described the first fossil remains assigned to Polyptychodon (see Hopson (2005) and Hopson, Wilkinson & Woods (2008), as well as critical comments by Wray & Gale (2006) and response by Hopson et al. (2006)). Still, Owen’s locality descriptions are often precise enough to allow for establishing approximate stratigraphic settings of the material in question.

The best documented oldest occurrence of Polyptychodon is most likely the name-bearing specimen of Polyptychodon continuus discovered in the “Kentish Rag Quarries near Maidstone” (Owen, 1841c). The strata of the alternating layers of sandy limestone (“ragstone”, “rag”) have long been known as a part of the Hythe Beds (Casey, 1961). Today, they represent the Hythe Formation of the Lower Greensand Group (Shand et al., 2003). In Kent, the formation ranges from the upper section of the lower Aptian to the lower portion of the upper Aptian (Casey, 1961; Shand et al., 2003). Thus, the oldest material ascribed to Polyptychodon is probably best considered as of late early Aptian age (∼120 Ma).

The most numerous finds attributed to Polyptychodon were described by Owen (1851) and Seeley (1869) from the “middle and lower Chalk or Chalk-marl” and “Cambridge Upper Greensand”, respectively, of Cambridgeshire. Even though precise stratigraphy is rather difficult to infer from the published data, it seems likely that most of the material originates from the Cambridge Greensand Member. The latter unit represents the lowermost portion of the West Melbury Marly Chalk Formation (a unit approximately equivalent to the “Chalk Marl” of the traditional scheme; see e.g., Hopson, 2005; p. 16). Although being deposited at the very beginning of the Cenomanian transgression, the Cambridge Greensand Member contains a macrofossil assemblage that is hypothesized to be largely reworked from the underlying Gault Formation (e.g., Hart, 1973; Hopson, 2005; Martill & Unwin, 2012). Thus, the age of most of the vertebrate material from this unit is probably late Albian (∼105 Ma), although it cannot be excluded that some rare finds are already of earliest Cenomanian age.

The youngest specimens assigned to Polyptychodon were discovered in the strata belonging to the Chalk Group but their exact provenances are often problematic. Possibly the youngest and best documented record classified as Polyptychodon is the specimen CAMSM B 75741 (see ‘Descriptions’ below). This tooth crown was unearthed from the uppermost lower Coniacian to the uppermost middle Santonian succession of the Seaford Chalk Formation of Gravesend, Kent (minimum age ∼84 Ma). In summary, the stratigraphic range of the English material classified as Polyptychodon is possibly as wide as late early Aptian to middle Santonian (∼120–∼84 Ma).

Descriptions

The character of the studied collection does not suggest any clear taxonomic distinctions. Thus, particular teeth are described separately according to their age and localities of discovery. The catalog numbers of the studied specimens together with their provenances are summarized in Table 2.

Table 2 The list of the material attributed to Polyptychodon examined in the present study.

Material	Catalog numbers	First mentioned in	Locality	Lithostratigraphy	Stage	
Two partial tooth crowns	CAMSM TN 3770.1.1 and TN 3770.2.1	This study	Folkestone (Kent)	Gault Formation	Middle to upper Albian	
119 teeth	CAMSM B 57282–382, B 57384, B 57400, B 57407–412, B 57852, TN 1716.1–9	Owen (1851; pp. 55–56), Seeley (1869; pp. xviii, 45), this study	Cambridge (Cambridgeshire)	Cambridge Greensand Member of the West Melbury Marly Chalk Formation	Lower Cenomaniana	
Two almost complete tooth crowns	CAMSM B 74968 and B 74969	This study	Hauxton (Cambridgeshire)	West Melbury Marly Chalk Formation and basal Zig Zag Chalk Formation	Lower to middle Cenomanian	
Three almost complete tooth crowns	CAMSM B 20624, B 20625, and B 20626	Seeley (1869; p. 3)	Cherry Hinton (Cambridgeshire)	West Melbury Marly Chalk Formation or the Zig Zag Chalk Formation?	Cenomanian	
One tooth crown and two roots	CAMSM B 75753, B 75754, and B 75755	This study	Haslingfield (Cambridgeshire)	West Melbury Marly Chalk Formation or the Zig Zag Chalk Formation	Cenomanian	
Three tooth crowns	CAMSM B 20619, B 20620, and B 20621	Seeley (1869; p. 3)	Halling (Kent)	Plenus Marls Member of the Holywell Nodular Chalk Formation?	Upper Cenomanian	
One tooth crown	CAMSM B 75741	This study	Gravesend (Kent)	Seaford Chalk Formation	Lower Coniacian to middle Santonian	
Two tooth crowns	CAMSM B 20622 and B 20623	Seeley (1869; p. 3)	Offhamb (Kent)	Chalk Group	? (Upper Cretaceous)	
Notes.

a Majority of the material from the Cambridge Greensand Member is derived from the upper Albian Gault Formation.

b The provenance might be incorrect (see text).

Middle to upper Albian of Folkestone

Material. Two partial tooth crowns (CAMSM TN 3770.1.1 and TN 3770.2.1).

Locality and age. Gault Formation, Folkestone, Kent. CAMSM TN 3770.1.1 was supposedly discovered in bed VII that approximately corresponds with the Euhoplites lautus Zone; upper part of the middle Albian (e.g., Knight & Morris, 1996; Owen, 2012). The information on the precise stratigraphic provenance of CAMSM TN 3770.2.1 is not available. It falls within the range of the Gault Formation at Folkestone (i.e., middle to upper Albian; Owen, 2012).

Description. Both tooth crowns are incomplete. Only a mesiolabial part of CAMSM TN 3770.1.1 (CH = ∼50 mm; Fig. 3A) is preserved, while CAMSM TN 3770.2.1 (CH = ∼35 mm; Fig. 3B) is accessible only linguodistally. It is impossible to precisely infer WLR in both tooth crowns. However, the exposed part of CAMSM TN 3770.2.1 suggests it was approaching 1. The preserved parts of both tooth crowns indicate that they were only slightly curved linguodistally.

Figure 3 Teeth from the Gault Formation.

Two partial teeth from the Gault Formation, Folkestone, Kent: (A) CAMSM TN 3770.1.1 from the labial view and (B) TN 3770.2.1 from the linguodistal? view. Scale bar = 1 cm.

Apicobasal ridges. Due to the fact that different parts are accessible, it is impossible to compare the extent and development of the apicobasal ridges of the tooth crowns. In CAMSM TN 3770.1.1, the mesiolabial face is almost unridged with probably three mesiolabially positioned ridges running through its whole apicobasal length. Basally, a few short and rather scattered ridges can be noticed. In CAMSM TN 3770.2.1, only the lingually to linguodistally positioned ridges are accessible. The ridges are well pronounced, relatively closely-spaced, and majority of them run through whole apicobasal length, with slightly shorter ridges being present between them.

Enamel surface. In CAMSM TN 3770.1.1, the enamel surface is rather smooth mesiolabially with very slight roughening on the basal one-third of the tooth crown. The enamel surface of CAMSM TN 3770.2.1 cannot be assessed because it is not enough exposed between the linguodistally positioned apicobasal ridges.

Late Albian material in the lower Cenomanian of Cambridge

Material. 119 teeth in various states of preservation (CAMSM B 57282–382, B 57384, B 57400, B 57407–412, B 57852, TN 1716.1–9).

Locality and age. Cambridgeshire. All studied teeth likely represent derivative material from the upper Albian of the Gault Formation and belong to the Cambridge Greensand Member of the West Melbury Marly Chalk Formation, deposited at early phases of the Cenomanian (e.g., Hopson, 2005; Martill & Unwin, 2012). Based on the co-occurring ammonite phosphatic moulds, this material may be assigned to the lower part of the traditional upper upper Albian Stoliczkaia dispar Zone (e.g., Cooper & Kennedy, 1977). Following modern subdivision of the upper Albian, these strata belong to the Mortoniceras fallax and M. perinflatum zones, with most material originating from the former (see e.g., Amédro, 2008).

Description. The material clearly consists of teeth from different jaw positions (large, moderately curved teeth, as well as small, probably posteriormost dentary tooth crowns; see ‘Discussion’ for information on the tooth row variability) undoubtedly belonging to multiple individuals of various ontogenetic stages. The tooth crowns are generally suboval to subcircular in cross-section, with WLR between 0.8 and 1 (median = ∼0.9). The color of the teeth varies from rather light to dark brown (e.g., CAMSM B 57408 and B 57338, respectively) with no clear boundary.

Apicobasal ridges. All tooth crowns possess clear apicobasally oriented ridges on their entire surfaces that are especially well pronounced linguodistally. The development of the ridges on the mesiolabial half of the tooth crowns, however, varies considerably. The differences are apparent especially in the lengths and density of the ridges (see Fig. 4). In some tooth crowns, the mesiolabially developed apicobasal ridges run only a few millimeters from the cervix dentis (e.g., CAMSM B 57378, B 57411). Other crowns, however, are ridged through their whole apicobasal height, reaching the apex or interrupting just a few millimeters below it (e.g., CAMSM B 57400, B 57341). Still, even though some tooth crowns possess clearly shorter mesiolabially positioned ridges, it is rather difficult to set a sharp boundary between such teeth and those with ridges developed almost up to the apex because the assemblage includes numerous “transitional” specimens (see also discussion on variability among the collection below). In most assessable teeth it is possible to identify only a few, possibly around six, apicobasal ridges that run continuously from the cervix dentis up to the apex. In those teeth with half flat or almost completely flat mesiolabial surfaces, this part of the enamel is bordered by two such ridges, a mesial and a labial one, with additional, mesiolabial ridge, separating the flat surface (see Fig. 5).

Figure 4 Distribution of apicobasal ridges.

Differences in the distribution and extent of mesiolabially positioned apicobasal ridges in selected tooth crowns from the Cambridge Greensand Member collection: (A) CAMSM B 57378 with almost flat mesiolabial face, (B) B 57341 with apicobasal ridges reaching the apex (C) B 57380 with very closely-spaced apicobasal ridges, and (D) B 57292 with distantly-spaced apicobasal ridges. All tooth crowns pictured from the labial view. Scale bar = 1 cm.

Figure 5 Possible taxonomy-relevant pattern in apicobasal ridges.

Possible taxonomy-relevant pattern in the mesially and mesiolabially positioned apicobasal ridges (marked 1, 2, and 3) in selected tooth crowns: (A) CAMSM TN 3770.1.1 from the Gault Formation and (B) CAMSM B 57411, and (C, D) B 57378 from the contemporary Cambridge Greensand Member collection. (A–C) pictured from the labial view; (D) pictured from the apical view. Scale bar = 1 cm.

Differences can also be observed in the density of ridges. Some tooth crowns possess rather distantly-spaced apicobasal ridges that are nearly constantly remote from each other throughout whole tooth crowns (this especially applies for smaller teeth). In others, then, the ridges are very closely-spaced (e.g., CAMSM B 57380; Fig. 4C). Unfortunately, no such tooth with densely ridged enamel preserves its apex.

Enamel surface. When the density of the apicobasal ridges is lower (often on the mesiolabial faces of the tooth crowns), the enamel surface is slightly roughened (see e.g., Fig. 4A).

Lower to middle Cenomanian of Hauxton

Material. Two almost complete tooth crowns with worn apices (CAMSM B 74968 and B 74969).

Locality and age. Hauxton, Cambridgeshire. Both teeth are described as being from the “Chalk Marl”. Following the current lithostratigraphic terminology (Hopson, 2005), this informal term approximately corresponds with the West Melbury Marly Chalk Formation and basal Zig Zag Chalk Formation. In terms of ammonite biostratigraphy, this interval ranges from the Mantelliceras mantelli to Acanthoceras rhotomagense zones (lower to lower middle Cenomanian) (Kennedy, 1969; Hopson, 2005).

Description. Both tooth crowns are almost complete, lacking only their apices which were clearly worn off. Both tooth crowns are slightly curved linguodistally and very similar in terms of their size and morphology (CAMSM B 74968: CH = ∼45 mm, WLR = 0.94; B 74969: CH = ∼50 mm, WLR = 0.92).

Apicobasal ridges. Mesiolabially, both teeth possess three clearly visible, rather pronounced, apicobasal ridges that are separated by half flat surfaces (see Figs. 6A and 6B). Linguodistally, both teeth have well pronounced and closely-spaced ridges that cover basal two-thirds of the tooth crowns. Only a few ridges reach the apices.

Enamel surface. Both teeth have roughened surfaces forming slightly vermicular striae between apicobasal ridges on their mesiolabial faces.

Lower to upper Cenomanian of Cherry Hinton

Material. Three almost complete tooth crowns (CAMSM B 20624, B 20625, and B 20626).

Locality and age. Cherry Hinton, Cambridgeshire. The material is labeled as originating from the “Lower Chalk”. This suggests that it was discovered in the strata belonging to the Gray Chalk Subgroup, namely the West Melbury Marly Chalk Formation or the Zig Zag Chalk Formation. Biostratigraphically, the “Lower Chalk” ranges from the Mantelliceras mantelli to Metoicoceras geslinianum ammonite zones; i.e., lower to upper Cenomanian (Kennedy, 1969; Hopson, 2005).

Figure 6 Teeth from the Chalk Group.

The tooth crowns from the “Chalk” ascribed to Polyptychodon: (A) CAMSM B 74968 [LV] and (B) B 74969 [LV] from the Hauxton, Cambridgeshire; (C) CAMSM B 20624 [LV], (D) B 20625 [LV], and (E) B 20626 [LV] from the Cherry Hinton, Cambridgeshire; (F) CAMSM B 75754 [LV] from the Haslingfield, Cambridgeshire; (G) CAMSM B 20619 [LV], (H) B 20621 [LV], and (I) B 20620 [LV?] from the Halling, Kent; (J) CAMSM B 75741 [LV] from the Gravesend, Kent; and (K) CAMSM B 20622 [LGV] and (L) B 20623 [LGV] from the Offham, Kent. LV, labial view; LGV, lingual view.

Description. All three teeth are small (CAMSM B 20624: CH = ∼30 mm, WLR = ∼0.88; B 20625: CH = ∼20 mm, WLR = ∼0.85; B 20626: CH = ∼17 mm, WLR = ∼0.9). CAMSM B 20624 and B 20625 are very slightly curved distally while B 20626 is slightly curved linguodistally (see Figs. 6C–6E). All teeth are almost complete. CAMSM B 20624 lacks its apex and a distobasal part, B 20625 lacks a small mesiobasal part, and B 20626 has its apex broken off.

Apicobasal ridges. CAMSM B 20624 and B 20625 have very well pronounced apicobasal ridges. Their distribution, however, is uneven. Mesiolabially and mesiolingually, the ridges are more distantly-spaced and terminate in the two-thirds of the crown heights. Mesially, CAMSM B 20624 has a single ridge running along the entire crown height. The apical half of the ridge has very irregular prominences accompanied by wrinkled enamel surface. The apicobasal ridges of CAMSM B 20626 are very fine and more closely-spaced. Due to the missing apex, the extent of the apicobasal ridges in CAMSM B 20626 cannot be fully assessed.

Enamel surface. The enamel surface is exposed along the whole circumference on the apical one-thirds of CAMSM B 20624 and B 20625. Except for the slight wrinkles mentioned above, the surface is smooth.

Lower to upper Cenomanian of Haslingfield

Material. One tooth crown and two roots (CAMSM B 75753, B 75754, and B 75755).

Locality and age. Haslingfield, Cambridgeshire. Labeled as being from the “Lower Chalk”, the teeth were likely discovered in the strata of the West Melbury Marly Chalk Formation or the Zig Zag Chalk Formation; Mantelliceras mantelli to Metoicoceras geslinianum zones; i.e., lower to upper Cenomanian (Kennedy, 1969; Hopson, 2005).

Description. The material from Haslingfield is reportedly associated but only a single tooth crown (CAMSM B 75754) can be assessed (Fig. 6F). The tooth crown is among the largest assessed crowns attributed to Polyptychodon (CH = ∼95 mm). It is curved linguodistally and suboval in cross-section (WLR = ∼0.9).

Apicobasal ridges. The apicobasal ridges are well pronounced and present around the entire circumference. Although an apicomesial part of the enamel is missing, this side of the tooth crown was clearly ridged only on its basal one-third. However, unlike in the case of CAMSM TN 3770.1.1 or some tooth crowns from among the Cambridge Greensand Member collection (see Fig. 5), which are similar in that their mesiolabial faces are largely unridged and the enamel surfaces are mostly exposed, none of the assessable apicobasal ridges is reaching the apex.

Enamel surface. From among the tooth crowns associated with P. interruptus, CAMSM B 75754 stands out with its rough enamel surface forming vermicular striae (see Fig. 6F and Fig. 7 for close-up).

Figure 7 Exposed enamel surface with vermicular striae.

Exposed enamel surface with well pronounced vermicular striae between adjacent apicobasal ridges (indicated by arrows) on the labial face of a tooth crown (CAMSM B 75754) from the lower to upper Cenomanian of Haslingfield.

Upper Cenomanian of Halling

Material. Three tooth crowns (CAMSM B 20619, B 20620, and B 20621).

Locality and age. Halling, Kent. The age of the material is somewhat problematic as the teeth are only labeled as being unearthed from the “Chalk”. The provided locality (Halling) might suggest several sites stratigraphically ranging from the “Lower” to the “Upper Chalk” of the old scheme. If the teeth originate from the Halling Chalk Pit, it would likely place them within the range of the Plenus Marls Member which is a basal unit at the Holywell Nodular Chalk Formation. Biostratigraphically, the Plenus Marls Member falls within the upper Cenomanian Metoicoceras geslinianum Zone (Hopson, 2005; Gale et al., 2005). Still, the suggested range must be considered with caution.

Description. Only the labial side of CAMSM B 20619 is assessable. The tooth crown is almost complete, lacking only an apicolabial part. It is slightly curved linguodistally and supposedly suboval in cross-section (CH = ∼30 mm, WLR = 0.9; Fig. 6G). Although only the basal half of B 20621 is preserved, the tooth crown is similar to B 20619 in the size (CH = ∼20 mm; Fig. 6H) and the distribution of the apicobasal ridges. Considering its cross-section, however, CAMSM B 20621 seems to be more labiolingually compressed (WLR = ∼0.75), although the precise value of WLR cannot be measured in either of the two tooth crowns as they are both partially preserved in matrix. CAMSM B 20620 differs significantly from B 20619 and B 20621 (Fig. 6I). Although being of comparable size (CH = ∼33 mm), CAMSM B 20620 is much slender, subcircular in cross-section (WLR = ∼1), and more distinctly curved distally.

Apicobasal ridges. The development of the apicobasal ridges of CAMSM B 20619 and 20620 is almost indistinguishable. In both tooth crowns the ridges are rather pronounced, developed around the entire circumferences of the crowns, and likely reach at least two-thirds of the crown heights. In CAMSM B 20619, however, the ridges seem to be more distantly-spaced. The apicobasal ridges of CAMSM B 20620 are much less pronounced, distantly-spaced, and of irregular extent. Linguo- and labiodistally, the apicobasal ridges are developed along almost whole crown height. Mesiolingually and mesiolabially, however, the ridges are present only on the basal one-third to basal half of the crown.

Enamel surface. Except for a small apicomesial part of the CAMSM B 20619, the enamel surface is not exposed in CAMSM B 20619 and B 20621. In CAMSM B 20620, it is smooth.

Lower Coniacian to middle Santonian of Gravesend

Material. One tooth crown (CAMSM B 75741).

Locality and age. Gravesend, Kent. The tooth crown is labeled as originating from the “[Micraster] coranguinum Zone” of the “Upper Chalk”. This suggests the tooth was likely unearthed from the uppermost lower Coniacian to the uppermost middle Santonian of the Seaford Chalk Formation (Hopson, 2005).

Description. Only the basal two-thirds of CAMSM B 75741 are present (CH = ∼45 mm; Fig. 6J). The apical part is worn off. The tooth crown is moderately curved linguodistally and “subhedral” in cross-section (WLR = ∼0.95).

Apicobasal ridges. Due to the absent apex, it is impossible to fully assess the extent of the apicobasal ridges (see Fig. 6J). However, the ridges are well pronounced and rather distantly-spaced around the entire circumference, with slightly increased density linguodistally.

Enamel surface. The enamel surfaces can be assessed only between adjacent apicobasal ridges on the mesiolabial side of the tooth crown. It is rough, forming vermicular striae.

Upper Cretaceous of Offham(?)

Material. Two tooth crowns (CAMSM B 20622 and B 20623).

Locality and age. Offham(?), Kent. The tooth crowns are only labeled as being discovered in the “Chalk”. However, the provenance does not seem to be correct. Offham is situated on the Aptian Hythe Formation (Dines et al., 1969), with the nearest chalk outcrop being around 3.5 km to the northwest of Offham village near Trottiscliffe (P Hopson, pers. comm., 2015). Since the specimens are preserved in chalk, the stratigraphic range might only be limited to Upper Cretaceous.

Description. The apical one-third of CAMSM B 20622 (CH = ∼40 mm) is absent (Fig. 6K). The tooth crown is slightly curved linguodistally and suboval in cross-section (WLR = ∼0.85). Only a lingual part of CAMSM B 20623 (CH = ∼35 mm) is accessible (Fig. 6L). The tooth crown seems moderately curved linguodistally. The WLR cannot be assessed due to a large part of the crown being absent.

Apicobasal ridges. The apicobasal ridges are less pronounced and very closely-spaced around the entire circumference of CAMSM B 20622 and on the preserved part of B 20623. There are no apparent flat surfaces on the mesiolabial part of CAMSM B 20622. Mesiolabially, however, the crown is partially covered by matrix and lacks the apex. Due to the missing parts, it is also impossible to fully assess the extent of the ridges though in CAMSM B 20622 most of them were clearly reaching more than two-thirds of the crown height. Although the apicalmost part of B 20623 is broken off, it seems that only a few of the ridges extended up the apex.

Enamel surface. The enamel surface is not visibly exposed between the apicobasal ridges in either of the tooth crowns.

Discussion

The phylogenetic position and taxonomic validity of Polyptychodon

Originally, Owen could not confidently assign Polyptychodon to any of the groups he recognized and listed it as Sauria incertae sedis (Owen, 1841c; p. 156). In his later writings, following the discovery of a partial lower jaw with one well-preserved tooth, Polyptychodon was “proved to be a ‘thecodont’ saurian” (Owen, 1850; p. 379) and generally associated with Crocodilia (Owen, 1850; Owen, 1851). Owen (1851) pointed out that some of the aforementioned fragmentary postcranial elements from Hythe, he discussed in 1841 and described ten years later, “approach somewhat to the Plesiosaurian type” (p. 52), but this statement did not alter his classification. Most recently, the material from Hythe was identified as an indeterminate macronarian sauropod (Mannion et al., 2013).

The taxonomic interpretation of Polyptychodon changed after the discovery of the incomplete pliosaurid skull from the “Lower Chalk” at Dorking (Owen, 1860; Owen, 1861). According to Owen (1861), these fossils, as well as three additional vertebrae pictured on his Tables V and VI, demonstrated that Polyptychodon interruptus is a sauropterygian rather than a crocodile.

Although the affiliation of Polyptychodon with plesiosaurs became well supported, its taxonomic validity was considered doubtful. Welles (1962), who attempted to review the taxonomy of Cretaceous plesiosaurs, concluded that both, P. continuus and P. interruptus, are nomina vana that “should be dropped”. However, his discussion on P. interruptus is problematic. Welles (1962; p. 61) correctly noted that the taxon was based on a single tooth that was illustrated but not described, and that no locality of its discovery was given. However, Owen (1850; p. 378) does not “credit Mr. Mackeson with finding it in the Greensand near Hythe” (contra Welles, 1962). It is also doubtful whether Owen really “refigures” the specimen in 1851 as Welles (1962; p. 61) noted. The tooth crown regarded by Welles (1962) to be the name-bearer of P. interruptus that Owen (1851; Table XIV, Figs. 1 and 2) illustrated is certainly very similar to the tooth crown pictured by Owen (1841b; Pl. 72, Fig. 4) in its overall shape and even its broken surfaces. However, the tooth crowns slightly differ in their basal shapes and in the distribution of the apicobasal ridges on their mesiolabial faces. The ridges in the tooth illustrated by Owen (1851; Table. XIV, Figs. 1 and 2) seem to be more closely-spaced and differently curved. Welles (1962) also listed P. interruptus among “Senonian” (a vague term that approximately unites Coniacian to Maastrichtian stages of the official stratigraphic scheme) plesiosaurs.

A year later, Welles & Slaughter (1963) described the species Polyptychodon hudsoni (Welles & Slaughter, 1963) (SMU 60313) and stated that “[t]he skull fragment [of P. hudsoni] preserved is almost the same as that of Polyptychodon interruptus figured by Owen (1861, pl. 4, Fig. 1) (i.e., the Dorking specimen; DOKDM G/1–2)” and that “[t]his specimen should be considered the type of the species since earlier descriptions were on teeth alone” (p. 131–132). Yet, this notion is based on an incorrect assumption that teeth de facto cannot possess autapomorphic features. Also, the passages from Welles & Slaughter (1963) cannot be considered as a valid neotype designation in accordance with ICZN (1999; Art. 75) due to the following reasons:

(1) The statement that earlier descriptions were based only on teeth, and therefore DOKDM G/1–2 should be the type, does not make it “designated with the express purpose of clarifying the taxonomic status or the type locality of a nominal taxon” (Art. 75.3.1).

(2) The study by Welles & Slaughter (1963) lacks “a statement of the characters that the author[s regard] as differentiating from other taxa the nominal species-group taxon for which the neotype is designated, or a bibliographic reference to such a statement” (Art. 75.3.2).

(3) It does not provide the “data and description sufficient to ensure recognition of the specimen designated” (Art. 75.3.3);

(4) and does not include “the authors’ reasons for believing the name-bearing type specimen(s) [...] to be lost or destroyed, and the steps that had been taken to trace it or them” (Art. 75.3.4).

(5) It also does not present any “evidence that the neotype is consistent with what is known of the former name-bearing type from the original description and from other sources [...]” (Art. 75.3.5).

(6) Further, it does not contain any “evidence that the neotype came as nearly as practicable from the original type locality [...] and, where relevant, from the same geological horizon or host species as the original name-bearing type [...]” (Art. 75.3.6).

(7) And finally, the publication by Welles & Slaughter (1963) does not include “a statement that the neotype is, or immediately upon publication has become, the property of a recognized scientific or educational institution, cited by name, that maintains a research collection, with proper facilities for preserving name-bearing types, and that makes them accessible for study” (Art. 75.3.6).

In their brief discussion on the systematic status of P. interruptus, Angst & Bardet (2016) noted that the name-bearing specimen of P. interruptus differs from Brachauchenius lucasi or Polyptychodon hudsoni in that it lacks branching “striations” (= apicobasal ridges). The lack of branching apicobasal ridges was also used as an argument for attribution of pliosaurid material (SDSM 34991, SDSM 35004–6) from the upper Cenomanian of the Greenhorn Limestone (South Dakota, USA) to P. interruptus (VonLoh & Bell, 1998). However, there does not seem to be a clear pattern in the distribution of this character. For example, among the Cambridge Greensand Member collection, CAMSM TN 1716.5 possesses at least three branching ridges that are adjacent to the cervix dentis linguodistally, while the teeth associated with the type specimen of P. hudsoni have clear branching ridges on their apical halves (D Madzia, pers. obs.). Similarly, uneven distribution of the apicobasal ridges was observed by Schumacher (2008) in the holotype of Megacephalosaurus eulerti (FHSM VP-321). Schumacher (2008; p. 215) noted that “branching striae are visibly present on several teeth, but that most striations do not branch” and that they “were observed near the base, middle, and tips of individual teeth and in various aspects (lateral, medial, etc.)”.

Angst & Bardet (2016) also seem to suggest that the only way to keep P. interruptus valid is by designing a neotype. Even though I was unable to identify any autapomorphies, or unique combination of characters, in the name-bearing specimen of P. interruptus, any attempt to designate DOKDM G/1–2 from the Cenomanian–middle Turonian at Dorking as the neotype of P. interruptus would currently be premature as the initial, and only, descriptions of this specimen by Owen (1860) and Owen (1861) are outdated. Nevertheless, DOKDM G/1–2 is now under revision and its anatomy and systematic affinities are to be covered in a separate paper.

Variability in the teeth attributed to Polyptychodon

The most important collection ascribed to Polyptychodon is undoubtedly the one from the Cambridge Greensand Member (hereafter, CGMc). It includes a high number of isolated tooth crowns (n = 119) clearly representing teeth from different jaw positions of several individuals. Although the precise age assignment of particular tooth specimens from among the CGMc is somewhat problematic due to the fact that the material was reworked, the potential time span to which it belongs (i.e., late Albian; predominantly from the range of the ammonite Mortoniceras fallax) is still narrow enough to enable its hypothetical assignment to a single taxon. To assess the amount of variability in the teeth classified as Polyptychodon, the tooth crown morphology can be assessed with respect to the variability in the dentitions of the most closely related pliosaurid taxa.

The knowledge of the variability in the tooth crown morphology relative to the position in jaws of Cretaceous pliosaurids is limited. The most extensive discussion was probably provided by Schumacher, Carpenter & Everhart (2013) for Megacephalosaurus eulerti. The maxillary teeth of M. eulerti are large through majority of the tooth row. Their size, however, gently decreases posteriorly. The dentary dentition, however, is morphologically more heterodont, with only the first five teeth being relatively large. The teeth from the following alveoli are gradually decreasing their size with no apparent abrupt changes. According to Schumacher, Carpenter & Everhart (2013; p. 622), similar condition is observable in the specimen MNA V9433 that is referable to Brachauchenius lucasi (Albright, Gillette & Titus, 2007).

The tooth morphology and variability is better studied for Late Jurassic pliosaurids (e.g., Taylor & Cruickshank, 1993; Sassoon, Noè & Benton, 2012; Benson et al., 2013; Sassoon, Foffa & Marek, 2015). Sassoon, Foffa & Marek (2015) noted that derived pliosaurids (e.g., Pliosaurus) are heterodont in terms of tooth size, shape, and regional partitioning. Two morphological “expansions” were noticed in the teeth located at the upper jaws, both of which followed by smaller and more curved tooth crowns. Similar dental anatomy was also observed in other Late Jurassic thalassophoneans (e.g., Taylor & Cruickshank, 1993; Sassoon, Noè & Benton, 2012; Benson et al., 2013). However, the variability in the development of the apicobasal ridges within particular tooth rows has not been evaluated yet.

Considering the currently known differences in the dental anatomy of the Late Jurassic and Cretaceous pliosaurids, brachauchenines, or at least their mid-Cretaceous members, seem to be less heterodont than their Jurassic relatives.

Figure 8 Pliosaurid tooth crown morphologies according to Tarlo (1960).

Tooth crown morphologies as observed in Callovian pliosaurids and pictured in labial view by Tarlo (1960): (A) Simolestes nowackianus, (B) Simolestes vorax, (C) Liopleurodon ferox, (D) Liopleurodon pachydeirus, (E) Pliosaurus andrewsi, and (F) Peloneustes philarchus. S. nowackianus (Oxfordian or Kimmeridgian of Ethiopia) currently represents a species of the teleosaurid thalattosuchian Machimosaurus, M. nowackianus (Bardet & Hua, 1996; Young et al., 2014) and the assignment of andrewsi to Pliosaurus seems unlikely (e.g., Benson et al., 2013). All other combinations are in use.

Comparisons to pliosaurid dental anatomy

The teeth of Cretaceous pliosaurids have not been studied in detail yet. Except for the branching apicobasal ridges discussed on several occasions (e.g., Schumacher, 2008; Schumacher, Carpenter & Everhart, 2013; Angst & Bardet, 2016; or above), the tooth crown morphology and enamel structural elements still remain to be assessed. In their cross-sections, the tooth crowns from among the CGMc show variability that would be consistent with that observable, for example, in the alveoli of the jaws of Brachauchenius lucasi (Angst & Bardet, 2016; Fig. 2). The cross-sections of the tooth crowns are generally of suboval to subcircular character. Some larger teeth, however, can also be characterized as very slightly “subhedral”.

The best comparative material is currently available for Late Jurassic species. Tarlo (1960) observed that the tooth crown morphology of Callovian pliosaurids (considered to be Oxfordian at that time (see Martill & Hudson, 1991; Hudson & Martill, 1994)) might be unique to species (see Fig. 8). Nevertheless, as the variability in the density and distribution of the apicobasal ridges in the tooth crowns from among the CGMc shows, some of these “Callovian morphotypes” might determine attribution to a particular species only when found in certain stratigraphic levels. For example, CGMc includes tooth crowns with almost flat mesiolabial faces (e.g., CAMSM B 57378; see Fig. 4). A similar morphology is observable in the specimen CAMSM TN 3770.1.1 (see Fig. 3) from the stratigraphically equivalent levels of the Gault Formation that is contemporary to the CGMc. Whereas these two specimens might still belong to closely related members of a single evolutionary lineage, or even be regarded the same species, a similarly-ridged tooth crown was also reported by Meyer (1856; Pl. II; Figs. 1–3) from the Oxfordian of the Canton of Aargau (Switzerland). Based on illustration alone, there are no substantial differences between this tooth (named “Ischyrodon meriani”) and CAMSM B 57378 from the CGMc or TN 3770.1.1 from the Gault Formation that could not be explained by individual variation when found in the same strata. On the other hand, Tarlo (1960) and Noè (2001) tentatively referred “Ischyrodon meriani” to Liopleurodon ferox.

Similarly, the CGMc includes specimens with ridges originating at the base of the tooth crowns and terminating in the middle or two-thirds of the tooth height, with only a few ridges reaching the apex. Such morphology was illustrated by Tarlo (1960) for Peloneustes philarchus and later described by Ketchum & Benson (2011).

The CGMc also consists of tooth crowns (e.g., CAMSM B 57380) with very closely-spaced apicobasal ridges around their entire circumferences. This morphology is strikingly different from the one with almost flat mesiolabial faces.

In addition to the density and extent of the apicobasal ridges, variability can also be observed in the development of these structures. In some teeth, the ridges are prominent. In others, on the other hand, the apicobasal ridges are less pronounced. Similar differences were illustrated by Tarlo (1960) for Jurassic taxa. Likewise, the enamel surface exposed between particular apicobasal ridges sometimes bear rough vermicular striae. Although it must be remembered that the CGMc is a reworked assemblage and, thus, the states of these structures might have been affected taphonomically in some teeth, it cannot be ruled out that these differences are at least partially taxonomy-dependent.

All tooth crowns from among the CGMc can be distinguished from the Kimmeridgian and Tithonian species of Pliosaurus based on their cross-sectional shape. The species of Pliosaurus have teeth with distinctive subtrihedral to trihedral cross-section (Knutsen, 2012; Benson et al., 2013). The teeth from among the CGMc, on the other hand, have suboval to subcircular cross-sections.

Are there any pathological teeth in the material attributed to Polyptychodon?

The probability that pathologies enhance the variability in the teeth attributed to Polyptychodon is considered negligible. The condition when the tooth crowns have almost flat mesiolabial faces (e.g., CAMSM TN 3770.1.1, CAMSM B 57378) is not regarded pathological as a possible taxonomy-relevant pattern is observed in the distribution and extent of apicobasal ridges of these crowns (see Fig. 5).

Figure 9 Pathological tooth crown.

Pathological tooth crown (CAMSM B 57333) from the Cambridge Greensand Member collection with unusually curved and interrupted apicobasal ridges on its mesiolabial side. Pictured in the labial view. Scale bar = 1 cm.

Among the studied material, several teeth have rather chaotically scattered apicobasal ridges on a very short segment around the circumference of the tooth crowns adjacent to the cervix dentis. However, this state does not seem to affect the overall distribution of the apicobasal ridges across the tooth crowns. The only significant example of pathological development in apicobasal ridges is proposed for CAMSM B 57333 from the CGMc. This tooth crown possesses an unusual configuration of abnormally curved ridges, resulting in that the mesiolabial face of the tooth crown is aberrantly exposed (see Fig. 9).

How many taxa form the studied tooth collection?

Until now, the studied dental material has been regarded as representing a single taxon, Polyptychodon interruptus. However, its considerable morphological variability, discussed above, suggests that the assemblage might be of multispecies character. It remains difficult to conclusively differentiate between particular morphological types that would support taxonomic distinctions because of the general lack of detailed studies on plesiosaur dentitions. Still, comparisons of certain tooth crowns suggest the material might belong to different taxa. For example, the specimens CAMSM B 57378, B 57341, and B 57380 from among the CGMc possibly represent teeth from similar jaw positions, yet the extent and frequency of their apicobasal ridges differ markedly (see Fig. 4).

Interestingly, CAMSM B 57378, and other tooth crowns with almost flat mesiolabial faces (e.g., CAMSM TN 3770.1.1 from the Gault Formation), bear some similarities to an isolated tooth crown from the upper Turonian of Dresden-Strehlen in Germany (FG 18/2010; Sachs et al., 2016) that was considered to be referable to Polycotylidae. In addition to the lack of shorter apicobasal ridges on the mesiolabial faces of the tooth crowns, CAMSM B 57378 and FG 18/2010 also share three prominent ridges on their mesiolabial surfaces running from the cervix dentis to the apex (D Madzia, pers. obs.; see also Sachs et al., 2016; Fig. 3B). Thus, if Sachs et al. (2016) identified the tooth-bearer correctly, it cannot be ruled out that a part of the studied dental material attributed to Polyptychodon belongs to polycotylids.

Although the teeth of polycotylid plesiosauroids are often slender and finely ridged (e.g., the species of Dolichorhynchops; Schmeisser McKean, 2012), the dentition of some taxa, such as Polycotylus latipinnis Cope, 1869 and Edgarosaurus muddi Druckenmiller, 2002, tends to be more robust in some regions of the tooth rows, thus resembling the anteriorly positioned teeth of brachauchenines.

Considering the above-mentioned, it is possible that CAMSM B 57378, B 57341, and B 57380 belong to different taxa or maybe even different larger clades. Unfortunately, an extensive and detailed assessment of the morphology and variability in the dentition within single jaws of particular robust-toothed plesiosaur taxa is needed prior to proposing any taxonomic decisions.

The impact of the new findings on the research of brachauchenine dentitions

Polyptychodon interruptus is a widely recognized taxon of a considerable historical value. However, it has never been a key component in the phylogenetic or paleobiogeographic studies of brachauchenine pliosaurids. Indeed, it has been emphasized that a revision of Polyptychodon is needed prior to inferring hypotheses that are to be based on its material (e.g., Sato et al., 2012).

Since the new findings show a substantial morphological variability in the tooth material ascribed to Polyptychodon, it is apparent that Cretaceous robust-toothed plesiosaurs were more diversified than usually assumed. Still, it would probably be premature to suggest a greater diversity for a particular clade, e.g., Brachaucheninae, as clear characters that enable to safely distinguish between the teeth of these pliosaurids and other groups of robust-toothed plesiosaurs, such as polycotylids, still remain to be established.

Also, brachauchenines did not necessarily constituted the only pliosaurid lineage that persisted to the Cretaceous. Zverkov (2015) reported on a single isolated tooth crown (h-216; Faculty of Geology, Lomonosov Moscow State University) from the Valanginian of Russia. Considering its trihedral cross-section, Zverkov (2015) concluded it might belong to an indeterminable species of Pliosaurus, a taxon lying outside Brachaucheninae. Such findings would indicate that at least two pliosaurid lineages crossed the Jurassic-Cretaceous boundary. Later the same year, Fischer et al. (2015) described a peculiar pliosaurid from the upper Hauterivian that they named Makhaira rossica. Similarly to h-216, the teeth of M. rossica were subtrihedral to trihedral in cross-sections. Following the results of their phylogenetic analysis, Fischer et al. (2015) suggested that Makhaira rossica represents the oldest known brachauchenine. Still, due to a low number of codable characters, it was emphasized to regard such phylogenetic position as tentative (Fischer et al., 2015). If additional research supports these findings, the evolutionary record of brachauchenines would be characterized, among others, by a transition in the shape of tooth crowns from subtrihedral or trihedral to suboval-subcircular.

Conclusions

Polyptychodon interruptus, the type species of Polyptychodon, was established on an isolated tooth crown that was possibly unearthed from the Upper Cretaceous strata of an unspecified locality in “Sussex”. This tooth crown is considered inaccessible and possibly lost. Still, its drawing by Owen (1841b) does not show any autapomorphies or unique combination of characters which makes P. interruptus a nomen dubium.

The reassessment of a representative sample of the material ascribed to Polyptychodon, that includes a part of the original specimens described by Richard Owen (e.g., Owen, 1851; Owen, 1860), shows a considerable variability in the tooth crown morphology and substantial differences in the age of particular specimens. The time span between the oldest and youngest record possibly exceeds 35 Ma (lower Aptian to ?middle Santonian). More importantly, still, the representative sample lacks any autapomorphic features that would enable the material, or a part of it, to be diagnosed. Rather, all observable characters seem to be widespread among pliosaurids and other robust-toothed plesiosaurs.

It is suggested that the material attributed to Polyptychodon represents a multispecies assemblage that possibly incorporates members of different plesiosaur clades (mostly pliosaurids but perhaps also polycotylids). The real taxonomic composition of these assemblages, however, cannot be presently identified with sufficient certainty.

As exemplified by the reinterpretation of the specimens from Japan and Argentina (Sato et al., 2012; O’Gorman & Varela, 2010; respectively), the referrals of plesiosaur material accompanied with robust teeth from numerous localities to Polyptychodon likely stemmed from the general absence of diagnosable pliosaurids and other robust-toothed plesiosaurs from the Cretaceous strata and, thus, relied merely on tradition rather than actual relationships.

From a global perspective, thus, Polyptychodon is a wastebasket taxon whose material originating from different localities should be reconsidered in separate studies.

I am indebted to Matt Riley and Dan Pemberton (both CAMSM) for access to the specimens in their care and their great help during my two stays in Cambridge. I also thank Marcin Machalski (Institute of Paleobiology, Polish Academy of Sciences) for helpful comments on the drafts of the MS and many interesting suggestions at different stages of the study, and Peter Hopson (British Geological Survey) for sharing his extensive knowledge and invaluable discussion on the Chalk Group. Sven Sachs (Naturkunde-Museum Bielefeld) kindly provided pictures of the teeth associated with “Polyptychodon” hudsoni and the specimen FG 18/2010. I am also indebted to Marian Dziewiński for taking pictures of the figured specimens and Magdalena Łukowiak (both Institute of Paleobiology, Polish Academy of Sciences) for her help with preparation of the figures. The reviews by Sven Sachs (Naturkunde-Museum Bielefeld) and Davide Foffa (University of Edinburgh) were very detailed and significantly improved the quality of the manuscript.

Institutional abbreviations

CAMSM Sedgwick Museum of Earth Sciences, University of Cambridge, Cambridge, UK

FG Geowissenschaftliche Sammlungen, Technische Universität Bergakademie Freiberg, Germany

FHSM Sternberg Museum of Natural History, Fort Hays State University, Hays, Kansas, USA

MNA Museum of Northern Arizona, Flagstaff, Arizona, USA

MNHN Muséum National d’Histoire Naturelle, Paris, France

NHMUK Natural History Museum, London, UK

SMU Schuler Museum of Paleontology, Southern Methodist University, Dallas, Texas, USA.

Other abbreviations

CH crown height, the distance between the most distal point at the base of the tooth crown and the crown apex

WLR width-to-length ratio, measured adjacent to the cervix dentis (the parameter is roughly equivalent to CBR sensu Smith, Vann & Dodson, 2005).

Additional Information and Declarations

Competing Interests

Author Contributions

Data Availability

The author declares there are no competing interests.

Daniel Madzia conceived and designed the experiments, performed the experiments, analyzed the data, contributed reagents/materials/analysis tools, wrote the paper, prepared figures and/or tables, reviewed drafts of the paper.

The following information was supplied regarding data availability:

The research in this article did not generate any raw data.

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
