# Peer review of "A reappraisal of Polyptychodon (Plesiosauria) from the Cretaceous of England"

_PeerJ, doi:10.7717/peerj.1998_

## Round 0.1 · original submission · Major Revisions

Dear authors,

As the two reviewers have not made a uniform recommendation, I have accepted the decision of 'major revision'. Please pay close attention to the recommendations made by reviewer two.

Once again, thank you for submitting your manuscript to PeerJ and I look forward to receiving your revision.

Reviewer 1 ·

Basic reporting

No Comments

Experimental design

No comments

Validity of the findings

No Comments

Additional comments

See attached document

Annotated reviews are not available for download in order to protect the identity of reviewers who chose to remain anonymous.

·

Basic reporting

The revision of Polyptychodon is a very much needed study and this manuscript constitutes a first organised attempt to tackle this controversy. The figures are well organised and beautifully represent the morphological variation described in the text.

Here below I attached a list of general comments that I think would improve the coherence and clarity of the study.

1) I feel that several key references should be discussed earlier in the manuscript.
This is ultimately a decision of the author, however I feel that in the current state the manuscript is presenting the information in an untidy way and it is hard to follow – in places (eg: the discussion of the genus should be presented earlier in the text).
In particular the ‘Introduction’ or the current “Historic background” would benefit of a paragraph explaining the history of the genus (as discussed in Angst & Bardet (2015). Welles & Slaughter (1963), Albright III, Gillette&Titus (2007a), Schumacher (2008) and Angst & Bardet (2015) should also be mentioned earlier.

2) Whilst very detailed, the description scheme of the teeth is not consistent throughout the text. I suggest that the author modify the descriptions by adopting the same scheme for all the teeth (eg: size, proportions, ornamentation (pattern, relief, distribution, density), wear?). The same section would benefit from further explanation of some of the terminology adopted (eg: 'vermiculated’, ‘roughening’, and others – see attached PDF). The same scheme and terminology should also be used consistently throughout the text – it is currently not the case as some teeth are described in more details than others.
(For more on terminology see for example Young et al. 2014, http://rsos.royalsocietypublishing.org/content/1/3/140269).
It may be worth considering adding a table identifying the main morphotypes and main features of crown in the collection - if possible.

3) More comparisons with brachauchenine and polycotylid dentition should be made either on a base-to-base case (eg. comparisons in the description sections), or they should be inserted (expanded) in the discussion. If the author has strong reasons for not including them in the first place, these should be mentioned in the text, otherwise additional discussion sections should be added to address this matter.

Experimental design

No comments

Validity of the findings

No comments

Additional comments

All the figure captions lack indication of the orientation of the tooth. Please add them in where possible. Further comments are listed below and divided by section.
ABSTRACT
Line 10-11. CConflicting may not be the best word to describe the interrelationship within Pliosauridae. As the author correctly underlines some taxa are currently problematic, but the main structure of the clade (paraphyletic Callovian pliosaurids and derived Brachaucheninae) is reasonably well supported. Benson et al. 2013 (PLOS – page 29) underline that the main problems are in the Kimmeridgian-Cretaceous part of the Pliosauridae tree (e.g.: politomy between Pliosaurus + Brachaucheninae). According to Benson et al. 2013 (PLOS – page 29), these are caused by wildcard taxa such as Pliosaurus irigensis and QM F51291. I suggest rephrasing the sentence to underline the taxonomic/sampling/undescribed material issues, rather than the phylogenetic one (which is direct consequence of them).
INTRODUCTION
Line 31-35. This paragraph is a little confusing as it is partially used to introduce plesiosaurians clades and partially to explain extinctions. I suggest that the author rephrase this and the next paragraph to address first groups and then extinctions at the Jr-K boundary.
Lines 37-39. As the author mentioned in the discussion, there is evidence of non-brachauchenine pliosaurids in the Cretaceous (Zverkov 2015). Reference to this work should be added in.
Lines 42-43. Poor taxic diversity in the Early Cretaceous should also be mentioned here (see Benson et al. 2010 – Proc R. Soc. B). Please also add a reference to Benson et al. 2013 in here.
Lines 56-60. Recently, Fischer et al. 2016 (Supplementary material) also described a pliosaurid tooth referred to Polyptychodon interruptus from Russia. Please add the reference and discuss it.
Line 70. Kimmeridigan (Late Jurassic).
Lines 70-79. Is there a reference number for the mentioned mandible? Is it the same figured by Lennier 1887? If not this and the other Polyptychodon specimens mentioned in Lennier 1887 should be added into the list.
Lines 125-126. Add “at the time” before the comma, or rephrase the sentence: “Many taxa introduced [..], due to the absence off [...].
HISTORICAL BACKGROUND
Line 129. Some key reference are missing here: Welles & Slaughter 1963, Albright III, Gillette&Titus, 2007a, Schumacher 2008, Angst & Bardet (2015)
Line 342. It should be mentioned that DOKMD G1/2 should be regarded as ‘Brachaunchenius indet.’ according to Benson et al. 2013 (PLOS – page 29)
DESCRIPTIONS
Line 410. In which group? Pliosauridae? Brachaucheninae, others? This statement has to be somehow supported. It would be helpful to less expert readers if the more subtle features of the teeth were highlighted in the figures (maybe with arrows).
Line 414. ‘Well developed’ is vague as it could refer to apicobasal development or length or relief. Please clarify here an through the text.
Lines 433-435. Here and in other sections the authors refers to “shape and morphology” to justify the position of a tooth in the tooth row. This is a bit vague. Please clearly state what features allows to identify a tooth as “anterior” or “posterior”. References are also needed here. Taylor & Cruickshank 1992 and Sassoon et al. 2015 described the tooth variation along the tooth row in pliosaurids. Schumacher et al 2013 also wrote about tooth variations in brachaucheninae. These references should be discussed here or elsewhere in the manuscript. This is a recurrent issue in the manuscript (eg. line 495), please add references or discuss crown variation along the tooth row separately.
Line 439. This is the only paragraph with sub-headings for different features of the tooth morphology. Please remove the headings or add it to all the other teeth. The description scheme should be kept consistent through the text.
Lines 442. “Mutual horizontal distance”. Is this concept the same to “density of the ornamentation” as mentioned in line 463? If so, please clarify it and use “density” for consistence. Change through the text (eg. line 458).
Line 459-461. “Some tooth crowns possess rather distantly-spaced apicobasal ridges that are nearly constantly remote from each other throughout whole tooth crowns (this especially applies for smaller, probably posterior teeth).”
Reference needed.
Lines 460-461. Please add a reference to a specimen and figure (eg. FIG3D) to clarify the differences in ornamentation.
Line 464. “vermicular”. Please explain this term.
Line 483. Can these “vermicular striae” be highlighted in the figures? (Ideally with a close up of it?). And has this pattern be described in the past? If so please change terminology or clarify the correspondence of the two.
Line 533. Two crown are called “almost indistinguishable”. Please elaborate the meaning of this or delete the sentence.
Line 539. “There is apparent no roughening”. Please explain this term. Does it mean that it lacks "vermiculations"?
Line 552. “the ridges are rather distantly-spaced” Regularly around the entire tooth crown?
Line 575. Shouldn't this section - or part of it - be included into "Historic background" or even constitute a section on its own in the introduction? This is ultimately the author decision, but I think that the explanation of the genus history would help to introduce the reasons why this study is needed and has been carried on.
Line 650. Additional comparisons should be added either here or on the descriptions. In particular Brachaucheninae teeth (and perhaps also polycotylids as well). Even if their descriptions are rare some of them are available in the literature (Slaughter & Welles 1963; Schumacher et al. 2013 etc..).
Line 662. “Oxfordian pliosaurids”. Most - if not all - "Oxfordian" pliosaurids described by Tarlo 1960 are now regarded as Callovian as they all come from Peterborough area (Peterborough Member - Oxford Clay Formation). See Martill & Hudson 1991, and change the discussion accordingly. As Polyptychodon teeth are compared to Callovian pliosauirds, I see no reason for not also including comparisons with the genus Pliosaurus.

---

## Round 0.2 · accepted · Accept

Dear author,

Following the reviewers recommendations, I am happy to accept your manuscript for publication.

Reviewer 1 ·

Basic reporting

No Comments

Experimental design

No Comments

Validity of the findings

No Comments

·

Basic reporting

I was very glad to see a new version of this manuscript.
I have to congratulate the author for the excellent revision work done. The author diligently addressed all the issues that me and the other reviewer pointed out in the previous version. All the comments and problems have been addressed or very good reasons were provided in other cases. In both cases I find the explanations clear, detailed and to the point.

In particular: the author addressed some nomenclature problems with the addition of further figures and modifications of the captions; the description section was greatly expanded and it is not more fluent and well organised; the discussion and introduction were also improved with additions on relevant literature and with more comparisons to existing materials.

I have no additional comments on the new paragraphs as they are well structured, well referenced and fit well in the manuscript body.

I look forward to see this manuscript published.

Experimental design

No comments

Validity of the findings

No comments

Additional comments

No comments